



# Insights into Lake Baikal Radiocarbon Age Offsets from a Database of AMS Radiocarbon Age Estimates

Samuel R. S. Newall[1], Anson W. Mackay[2], Natalia Piotrowska[3], Maarten Blaauw[4]

[1]Earth Sciences, UC Santa Barbara, Santa Barbara, 93106, USA
[2]Department of Geography, Environmental Change Research Centre, UCL, London, WC1E 6BT, England, UK
[3]Division of Geochronology and Environmental Isotopes, Institute of Physics – CSE, Silesian University of Technology, Konarskiego 22B, 44-100 Gliwice, Poland
[4]School of Natural and Built Environment, Queen's University Belfast, Belfast, BT7 1NN, Northern Ireland, UK

*Correspondence to:* Samuel R. S. Newall (newall@ucsb.edu)

**Abstract.** Radiocarbon dates are an essential tool for dating non-varved lake sediments, however their interpretation is hindered by issues such as reservoir age or contamination which culminate in age estimates that can be thousands of years younger or older than the true depositional age of that sediment (we call this an age offset). Often, precise estimators of the radiocarbon age offset are not available, as in the case of Lake Baikal. Linear regression of uncalibrated radiocarbon dates has been used to estimate the age offset, with answers ranging from 0 to 1500 $^{14}$C yr BP. These have been interpreted to suggest that different regions of Lake Baikal have different age offsets, although some dispute this. Other estimators have returned estimates of approximately 2000 $^{14}$C yr BP. Despite this, most previous studies have not included any estimates of uncertainties for these age offsets in their proxy analysis, or have included uncertainty of, at most, ± 90 $^{14}$C yr. Here, we present a complete database of published AMS radiocarbon dates from Lake Baikal sediment cores up to 2023 and, using this, review the use of linear regression on uncalibrated radiocarbon ages as a method for estimating age offsets from the sediments of Lake Baikal. We apply a standardised linear regression age offset method to all cores in our database to better quantify the age offset of Total Organic Carbon (TOC) in the lake's sediments. We conclude that there is no statistically significant evidence from linear regression methods for a large difference in age offset in different regions of Lake Baikal. Our results return a lake-wide age offset estimate of TOC of 1.56 ± 0.75 $^{14}$C kyr BP, suggesting previous studies in Lake Baikal have significantly underestimated the temporal uncertainty of radiocarbon ages. Finally, our results are a caution that linear regression-based age offset estimates in lake sediments have a large uncertainty that might only be observable with multiple datasets.

## 1 Introduction

Lake sediments are natural archives that contain information on environmental histories, spanning every continent, at timescales from the past few decades to tens of millions of years. Spatially therefore, lakes contain palaeoenvironmental information allowing space-time reconstructions of, for example, human (Dubois et al., 2018) and climate change impacts on the environment (Fritz, 2008). Reconstructing past environments from lake sediments requires appropriate dating techniques and chronology construction. Radiocarbon dating is one of the most common dating techniques, with an ~50,000-year range of applicability that includes the transition from the Last Glacial Maximum to the Holocene, one of the most studied periods of paleoclimate. The process of using radiocarbon dates includes offset correction, calibration, and age-depth modelling – all aspects that introduce temporal uncertainty, a significant but often ignored limitation to paleoclimate research (Snyder, 2010), possibly



because of the statistical complexity in propagating temporal uncertainty. Radiocarbon calibration and age-depth
modelling techniques are regularly improved and updated (Reimer, 2022), facilitating better understanding of
radiocarbon analyses and the opportunity to reduce temporal uncertainty, however this can be challenging if the
radiocarbon data is not easily findable or accessible. We present this database of accelerator mass spectrometry
(AMS) radiocarbon dates from Lake Baikal sediment cores to promote FAIR principles (Wilkinson et al., 2016)
and facilitate improvement of Lake Baikal paleoclimate reconstructions. We then use the database to perform
repeat estimates of the radiocarbon age offset of total organic carbon (TOC) in the lake's sediments, using a linear
regression method, to evaluate the most likely TOC age offset, its uncertainty, and whether the age offset differs
between the two most studied regions of the lake, Academician's Ridge and Buguldeika Saddle.

An often-overlooked aspect of temporal uncertainty in radiocarbon data is the age-offset, which we define as the
difference between the true age of a sample and the analysed age. We prefer the term age-offset to the more
commonly used term reservoir age, as the latter term may be interpreted differently among different groups. For
example, marine reservoir ages are conceptually linked to the idea of slow internal mixing of a body of water,
leading to aged water masses, and therefore corrections may be based on modelling/evaluating such circulation
(Stuiver et al., 1986): In Lake Baikal the ventilation time is known to be less than 25 years (Weiss et al., 1991),
which rules out the possibility of a large reservoir age associated with lake mixing processes. The term age-offset
encompasses a reservoir age induced by mixing and may also include: a hardwater effect due to the presence of
carbonate rocks in the lake; redeposition of older sediment; and systematic contamination; among other things.
Our use of the term may closely mirror the term freshwater reservoir effect, as is often used in archaeological
studies (Ascough et al., 2011; Philippsen, 2013; Schulting et al., 2022). When discussing a lake core, the age
offset therefore represents the difference between the time since a layer of sediment was deposited in the lake and
the radiocarbon age returned from analysing a given sample from that layer of sediment.

The presence of a significant age offset of TOC-based radiocarbon dates in Lake Baikal was confirmed by Colman
et al. (1996), who used a linear regression-based estimation method on a suite of cores. They found age offsets of
approximately 400 yr in Academician Ridge and approximately 1500 yr in Buguldeika Saddle, which they
suggested may be due to reworked sediment from the Selenga River outflowing near the Buguldeika Saddle.
Subsequent papers, have used a similar linear regression method (Demske et al., 2005; Karabanov et al., 2004),
or different methods such as: directly dating the surface sediment (Murakami et al., 2012); using the Younger
Dryas radiocarbon plateau as a tie-point (Watanabe et al., 2009a); comparing TOC ages to pollen or diatomaceous
sediment ages (Nara et al., 2010); using wood radiocarbon ages (Prokopenko et al., 2007); or equating it to the
residence time of the lake (Nara et al., 2023). The results range from 380 14C yr (Nara et al., 2023) to 2,100 +/-
90 14C yr (Watanabe et al., 2009a).

Despite the evident uncertainty in estimating the age offset of Lake Baikal, many papers do not use uncertain
estimates of age offset (e.g. Murakami et al., 2012; Nara et al., 2010, 2023; Prokopenko et al., 2007) and those
that do have very small uncertainty ranges (e.g. +/- 90 years; Watanabe et al., 2009). One reason for this in older
papers was likely that statistical packages to incorporate such offsets were not available or were not user friendly.
This is no longer the case (Sweeney et al., 2018). Bayesian age-depth modelling software are now more user-



friendly and sophisticated (i.e. Bacon, Bchron, and OxCal by Blaauw and Christen, 2011; Haslett and Parnell,
2008; and Ramsey, 2008 respectively) and the development of techniques to analyse the resulting age-uncertain
records has been prolific (i.e. Anchukaitis and Tierney, 2013; Franke and Donner, 2019; Hu et al., 2017;
McClelland et al., 2021; McKay et al., 2021; Rehfeld and Kurths, 2014).

One hurdle when using Bayesian models, however, is that many users do not know how large the uncertainties
they want to incorporate are. This is the problem we faced with radiocarbon age offsets when attempting to
construct age-depth models for Lake Baikal, with the literature clearly indicating a large range of estimated age
offsets but no advice regarding how to input such information into an age-depth modelling software. The goal of
this study is to collect all published AMS radiocarbon data from Lake Baikal and use a single method to estimate
age offset for all suitable cores. By making multiple estimates on different cores, we deliver an estimate of age-
offset with a calculated uncertainty estimate. We use a linear regression-based technique for our estimates, which
has been used by multiple studies of Lake Baikal (Colman et al., 1996, Demske et al., 2005, Karabanov et al.,
2004) and on the Tibetan Plateau (see discussion in Hou et al. 2012). Because of data limitations, we only perform
this analysis for radiocarbon dates from TOC, but published radiocarbon dates from other sample types are also
included in our dataset, which we make public to facilitate future research and so others can perform similar
analyses if desired.

A number of studies have curated regional radiocarbon datasets with the aim of developing more robust age-depth
models for their respective cores to allow their temporal uncertainty to be integrated into future interpretations of
their paleoclimate proxies (Giesecke et al., 2014; Goring et al., 2012; Wang et al., 2019; Zimmerman and Wahl,
2020).  As far as we are aware, no systematic study has applied such an approach to a single lake.

Lake Baikal is the oldest, deepest, and most voluminous lake in the world. The lake's surface area covers 23,000
km$^2$ while its catchment spans over 500,000 km$^2$, reaching into northern Mongolia. Sediments from Lake Baikal
have been used to reconstruct climate change as far back as the Miocene (Antipin et al., 2001; Williams et al.,
2001), although most studies have focussed on the later stages of the Quaternary period (Mackay et al., 2011,
2022; Prokopenko et al., 2001). Of relevance to this study, is the timeframe encompassed by radiocarbon dating,
a period of very marked rapid and abrupt climate change, that allows insights to be gained as to how ecosystems
respond to global warming at rates observed today. Given the size of Lake Baikal (c. 630 km long) with its
complex basin morphology linked to tectonic activity, sediment accumulation rates (SAR) vary widely depending
on sedimentary depositional environments (Charlet et al., 2005). Comparing sediment records from different
depositional environments has proven challenging in Baikal primarily due to uncertainties associated with
radiocarbon dating, including age offsets (Prokopenko et al., 2007). One key question that has arisen, and is
addressed in this study, is whether sediment from different parts of the lake experience different age offsets.

The principal aims of this study are to: (1) Compile all AMS radiocarbon data available for Lake Baikal up to the
end of 2023; (2) Estimate the TOC age offset for Lake Baikal, including an estimate of its uncertainty; (3)
Critically evaluate the use of the linear regression method in estimating age offsets.





## 2 Method

### 2.1 Dataset Collection

Collation of studies which have published and/or utilised radiocarbon dates from Lake Baikal sediments were undertaken initially using Google Scholar with search terms such as "Lake Baikal" and "radiocarbon" alongside "Palaeoclimate", "Paleoclimate", "Age Depth Modelling", "Holocene", "LGIT". Grey literature, especially reports published pre-1995 were also consulted, including those in Russian, English and Japanese. Research leads (identified from corresponding author status in publications) were also contacted. Articles were read and their citations and references interrogated, leading to ~80 relevant papers being identified. Although our approach did not set out to be a systematic review, the five basic steps required for a review were followed including (i) careful framing of the question, (ii) identification of relevant work, (iii) assessment of the quality of identified work, (iv) summarising the evidence and (v) interpretation of the findings (Khan et al., 2003).

Metadata and radiocarbon data was recorded for all cores with radiocarbon data identified from the literature. Each core was assigned to a region of the lake - as is common in Lake Baikal literature due to the lake's size and cores reported with differing names in the literature are reported under a single name.

### 2.2 Data Organisation

There are several data repositories that host radiocarbon data; however, none provided a sufficient radiocarbon data reporting standard for this study. The most relevant databases to paleolimnological research are Neotoma, Pangaea, NOAA and VARDA. Neotoma, Pangaea and VARDA do not have standardised radiocarbon data reporting templates. The NOAA paleoclimate data template was not suitable for using legacy data of varying standards. The depth fields used are 'depth_top' and 'depth_bottom' which does not allow for the input of data for which only a middle depth was reported. Additional data such as $\delta^{13}C$ or carbon yield measurements were also not included. This would not allow the full reporting of AMS radiocarbon data, as is the aim of this study. We created a standardised data reporting template focused on being suitable for radiocarbon analyses from lake sediments. Our reporting template draws heavily on Millard (2014) and Stuiver and Polach (1977).

Regarding data identification, we provide both the laboratory code (a requirement for radiocarbon dates) and the section code when possible. Regarding depth information, data found reported in the literature in the following ways: top depth and bottom depth; middle depth and thickness; only middle depth. To accommodate all this information, we provide the top, middle, bottom depth and thickness of the sample. Where cores had depth corrections (for known losses of top sediment) we provide a corrected depth for each sample. We provide the radiocarbon age as they were reported in the original papers, as conventional $^{14}C$ age (Stuiver and Polach, 1977), with the standard $1\sigma$ error. We also report $\delta^{13}C$ values and error when they were provided, specifying the method used for $\delta^{13}C$ evaluation. Carbon yield values are included when provided. Lastly the original references are provided, and any difference between the original data and the provided data is explained as a comment.

The selection of fields was driven by our focus on TOC. As we do not perform calibration on any of the dates, we do not provide any calibrated date or calibration information. Furthermore, the data format does not include

indication of whether an age was rejected, as rejection can vary across publications. We do not provide our own
indication of this because our analysis only includes a subset of the total data, hence would be an incomplete
quality assurance. All data should be carefully considered by a researcher before any reuse.

For each core, we provide the core name, the general region of the core within the lake (i.e. Buguldeika Saddle or
Academician Ridge), latitude and longitude in degrees, water depth of drilling site, coring method used, length of
the core, references for original data and comments describing any corrections to the data made by us or providing
explanation for depth correction.

### 2.3 Age Offset Estimation

As previously described, the most common approach to estimating age offset in Lake Baikal is using a linear
regression. This method assumes that the sedimentation rate at a given location has not varied through time. A
linear regression of radiocarbon age against depth down a sediment core is made, returning the slope (i.e. the best
fit sedimentation rate) and the y-intercept. The y-intercept value, which we term the "apparent surface age" (ASA),
is taken to be the age offset. It is important to note that this approach assumes the age offset is essentially constant
over the period included in the linear regression. Different studies have calculated this linear regression in different
ways, with the primary difference being how many ages they use in the calculation. For example, Colman et al.
(1996) used two linear regression methods, one which used only the top two dates of a core and one where all
dates younger than 13 $^{14}$C kyr BP were used. Karabanov et al. (2004) and Demske et al. (2005) also apply a linear
regression method to calculate age offset in their study but do not describe what subset of ages they used for each
regression. We follow Colman et al. (1996) in their creation of composite cores for cores they report from the
same drilling site.

To evaluate the similarity or spread of age offsets estimated with a linear regression throughout Lake Baikal, we
apply a single linear regression method to each (composite) core in Lake Baikal that is suitable (Figure 1). A
simple linear regression is performed on the mean radiocarbon ages of each (composite) core, for ages that are
younger than 13 $^{14}$C kyr BP, following Colman et al. (1996), against the median depth of those ages. Radiocarbon
ages are used instead of calibrated ages because calibration requires estimating the age offset. The radiocarbon
profile of each (composite) core was viewed beforehand to remove outliers and to check that the ages are generally
ageing with increasing depth and are approximately linear. Cores that do not follow this description are excluded
from this analysis.

The exclusion of ages older than 13 $^{14}$C kyr BP follows from the change in sediment type at approximately this
age, from diatom-poor glacial sediments to diatom-rich interglacial sediments. By choosing to do linear regression
on all ages younger than 13 $^{14}$C kyr BP, we make the most of the constancy of Lake Baikal's sedimentation rates,
whilst reducing the importance of this assumption by cutting off ages that are older than the age for which the
major glacial-interglacial shift in sediment composition is observed. Further, this shortens the time over which we
must assume a constant age offset and reduces the impact of the error of individual $^{14}$C ages, which increases for
older ages. Finally, it allows us direct comparison to the results of Colman et al. (1996).


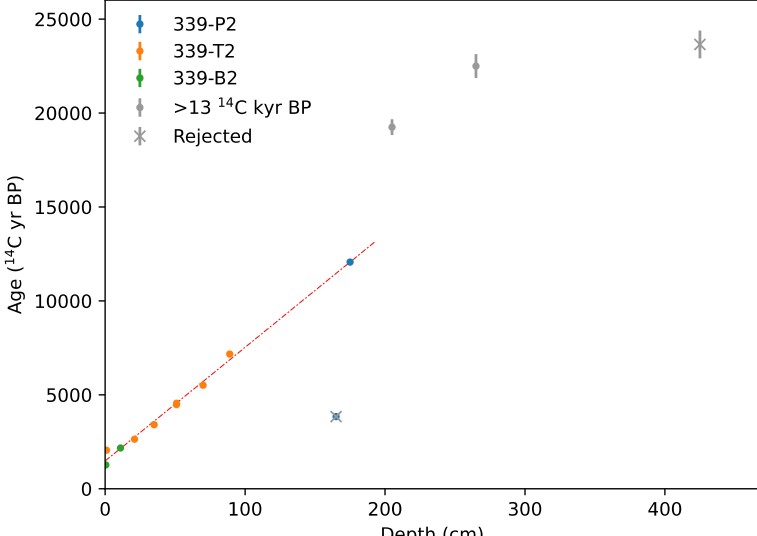


**Figure 1: An example of the creation of composite cores using cores from the same drilling site, following Colman et**
**al. (1996). The radiocarbon ages from three cores are plotted against depth, with circles showing the mean radiocarbon**
**age and bars showing the analytical uncertainty. The rejected ages follow the interpretation of Colman et al. (1996),**
**and those older than 13 $^{14}$C kyr BP are not used in the linear regression. The y-intercept, which we call the ASA, is 1.48**
**$^{14}$C kyr BP. In our interpretation of this core, we additionally rejected the 2$^{nd}$ deepest date from 339-P2 (the single blue**
**dot in this figure), because all other ages from this core are clearly problematic, but doing this also returns an ASA of**
**1.48 $^{14}$C kyr BP, so there is no impact of this on the results.**
**3 Results**
**3.1 Core Data Overview**
Our review identified 51 cores that contained AMS $^{14}$C dates, encompassing 518 radiocarbon datapoints (Table
1; Figure 2). All data either came from tables in the literature or personal communications from Fumiko Nara,
Takahiro Watanabe, Steven Colman, George Swann, Takuma Murakami and Masayo Minami. The cores are
mainly taken from two broad regions: the Academician Ridge, separating the Northern Basin and Central Basin,
and the Buguldeika Saddle, separating the Central Basin and the Southern Basin. Bathymetric highs such as these
are often chosen as coring sites because they often provide continuous and uninterrupted sediment records, unlike
slopes or basins. Some coring has been done in the deeper waters of each of the three basins, however these are
few.





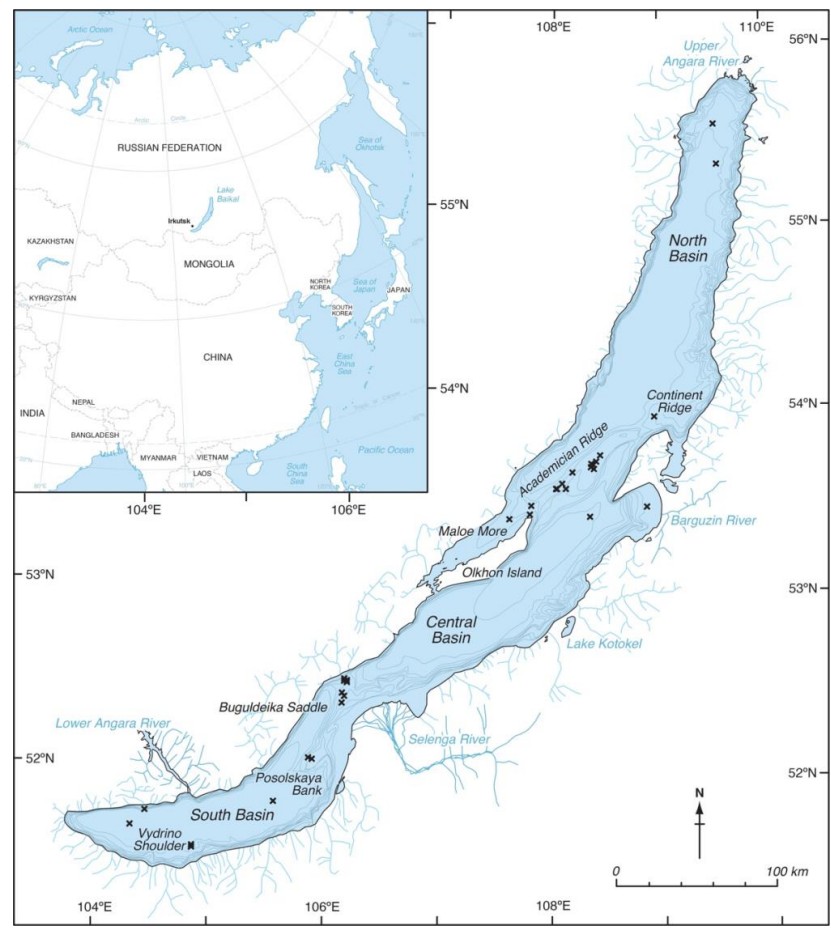


**Figure 2: Map of all cores in Lake Baikal, showing the groupings by location.**


| Region | Core Name | Latitude (º) | Longitude (º) | Depth (m) | References |
|---|---|---|---|---|---|
| **Academician Ridge** | 18-B1 | 53.56 | 108.01 | 345 | Colman et al. (1996) |
| | 18-P2 | 53.56 | 108.01 | 345 | Colman et al. (1996); Nakamura et al. (2003) |
| | 307-A3 | 53.59 | 108.07 | 335 | Colman et al. (1996) |
| | 331-P1 | 53.47 | 107.79 | 360 | Colman et al. (1996) |
| | 331-T1 | 53.47 | 107.79 | 360 | Colman et al. (1996) |
| | 333-P2 | 53.65 | 108.16 | 390 | Colman et al. (1996) |
| | 333-T2 | 53.65 | 108.16 | 390 | Colman et al. (1996) |
| | 340-B1 | 53.67 | 108.36 | 280 | Colman et al. (1996) |
| | 340-P1 | 53.67 | 108.36 | 280 | Colman et al. (1996) |
| | 340-T1 | 53.67 | 108.36 | 280 | Colman et al. (1996) |





| | BDP96-1 | 53.70 | 108.35 | 335 | Nakamura et al. (2003) |
|---|---|---|---|---|---|
| | BDP96-2 | 53.70 | 108.35 | 335 | Nakamura et al. (2003) |
| | BDP98-1 | 53.74 | 108.41 | 325 | Nakamura et al. (2003) |
| | VER98-1 St.5GC | 53.74 | 108.41 | 325 | Watanabe et al. (2009a); Watanabe et al. (2009b); Watanabe, personal communication |
| | VER98-1 St.5PC | 53.74 | 108.41 | 325 | Watanabe et al. (2009a); Watanabe, personal communication |
| | VER98-1 St.6GC | 53.69 | 108.35 | 335 | Watanabe et al. (2009a); Watanabe, personal communication |
| | Ver93-2 St.4-PC | 53.56 | 108.02 | 356 | Nakamura et al. (2003) |
| | Ver94-5 St.16-PC | 53.71 | 108.38 | 310 | Nakamura et al. (2003) |
| | Ver94-5 St.16-Pilot | 53.71 | 108.38 | 310 | Nakamura et al. (2003) |
| | Ver94-5 St.19-PC | 53.56 | 108.01 | 350 | Nakamura et al. (2003) |
| | Ver96-2 St.3-GC | 53.7 | 108.35 | 320 | Nakamura et al. (2003) |
| | Ver96-2 St.7-PC | 53.56 | 108.1 | * | Nakamura et al. (2003) |
| | Ver96-2 St.7-Pilot | 53.56 | 108.1 | * | Nakamura et al. (2003) |
| | Ver97-1 St.6 | 53.68 | 108.33 | 335 | Nakamura et al. (2003); Sakai (2006) |
| **Buguldeika Saddle** | 305-A5 | 52.4 | 106.12 | 290 | Colman et al. (1996) |
| | 316-P3 | 52.44 | 106.15 | 300 | Colman et al. (1996) |
| | 316-T3 | 52.44 | 106.15 | 300 | Colman et al. (1996) |
| | 339-B2 | 52.51 | 106.17 | 375 | Colman et al. (1996) |
| | 339-P2 | 52.52 | 106.17 | 375 | Colman et al. (1996) |
| | 339-T2 | 52.52 | 106.17 | 375 | Colman et al. (1996) |
| | BDP93-1 | 52.52 | 106.15 | 354 | Colman et al. (1996); Nakamura et al. (2003) |
| | BDP93-2 | 52.52 | 106.15 | 354 | Colman et al. (1996); Nakamura et al. (2003) |





|  | BSS06-G2 | 52.46 | 106.13 | 360 | Murakami et al. (2012) |
|---|---|---|---|---|---|
|  | VER93-2 St.24GC | 52.52 | 106.15 | 355 | Karabanov et al. (2004); Tarasov et al. (2007) |
|  | VER99G12 | 52.53 | 106.15 | 350 | Watanabe et al. (2007); Watanabe et al. (2009b); Nara et al. (2010) |
| **Barguzin Bay** | BarguzinBay18 | 53.42 | 108.82 | 41 | Fedotov et al. (2023) |
| **Central Basin** | 308-A3 | 53.42 | 108.32 | 1700 | Colman et al. (1996) |
| **Continent Ridge** | CON01-603-5 | 53.95 | 108.91 | 386 | Piotrowska et al. (2004); |
| **Maloe More** | 287-K2 | 53.42 | 107.78 | 300 | Colman et al. (1996) |
|  | 342-B1 | 53.4 | 107.59 | 240 | Colman et al. (1996) |
|  | 342-P1 | 53.4 | 107.59 | 240 | Colman et al. (1996) |
|  | 342-T1 | 53.4 | 107.59 | 240 | Colman et al. (1996) |
| **Northern Basin** | 323-PC1 | 55.54 | 109.52 | 710 | Ogura (1992); Nakamura et al. (2003); |
|  | Ver94-5 St.22-GC | 55.32 | 109.54 | 825 | Nakamura et al. (2003) |
| **Posolskoe Bank** | CON01-606-3 | 52.08 | 105.87 | 130 | Piotrowska et al. (2004) |
|  | Ver.99 G-6 | 52.09 | 105.84 | 200 | Tani et al. (2002) |
| **Southern Basin** | BAIK13-1C | 51.77 | 104.42 | 1360 | Swann et al. (2020) |
|  | BAIK13-4F | 51.69 | 104.3 | 1360 | Swann et al. (2020) |
|  | BDP97-1 | 51.85 | 105.55 | 1450 | Nakamura et al. (2003) |
| **Vydrino Shoulder** | CON01-605-3 | 51.59 | 104.85 | 675 | Demske et al. (2005) |
|  | CON01-605-5 | 51.58 | 104.85 | 665 | Piotrowska et al. (2004); Demske et al. (2005) |

**Table 1: A list of all cores for which radiocarbon data was found. Boxes with asterisks denote information that was not found.**

The location data provided for core CON01-603-5 by Piotrowska et al. (2004) and for core 287-K2 by placed the cores outside the boundaries of the lake. The location of 287-K2 was corrected by sight to match the locations provided on the map figures of Colman et al. (1996) and the location of CON01-603-5 was revised to fit that of Demske et al. (2005). Numerous slightly differing location data for BDP96-1 and BDP96-2 were found (Kashiwaya et al., 2001; Nakamura et al., 2003; Sakai et al., 2000), being 20 km apart at most. We use the value from Nakamura et al. (2003). Note, latitude/longitude data for core Ver97-1 St.6 was only found to the precision of degree minutes, not degree seconds (Sakai, 2006).





To aid the reader in finding the locations of cores in the densely cored regions, we provide higher resolution maps
of Academician Ridge and Buguldeika Saddle (Figure 3).

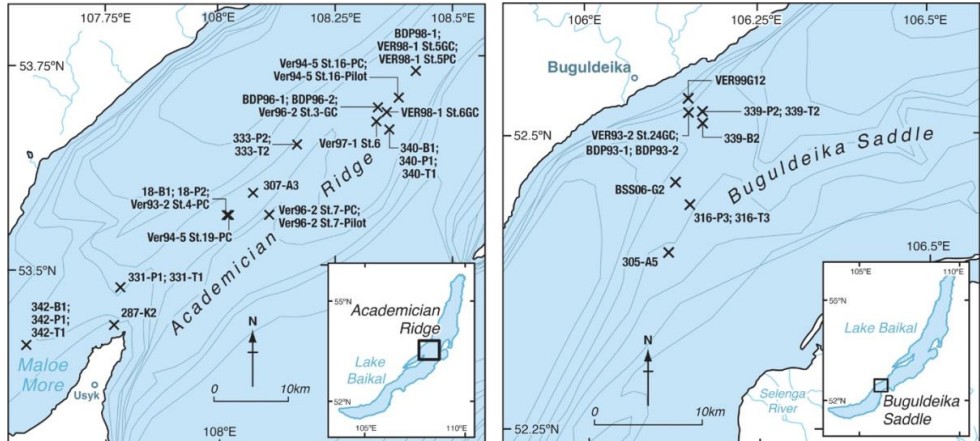

**Figure 3: Detailed maps of the core locations in Academician Ridge (left) and Buguldeika Saddle (right). Black crosses**
**denote core locations, some crosses represent multiple cores.**

## 3.2 Radiocarbon Ages Overview

Many cores have been analysed for radiocarbon ages in Lake Baikal, few have had many dates evaluated. The
cores' radiocarbon profiles are illustrated in Figure 4. The cores in the database have a mean of 10 dates per core,
but one core, VER99G12, has over 70 dates. Most radiocarbon dates (438 dates) in our collection are from TOC
Pollen concentrates have been used too (42 dates); however, they are not nearly as widely exploited due to their
more intensive preparatory workload. Further, the pollen concentrate dates still seem to suffer from age offsets,
as they show non-zero surface ages after regression. A few other materials have been dated but only in very low
numbers. These include total lipids (9 dates), picked organic matter (POM; 7 dates), fine organic matter (FOM; 5
dates); diatom/pelitic silt (5 dates), lipid fraction (2 dates) and wood (2 dates). Note that POM and FOM relate to
two different styles of sampling TOC, used in Colman et al. (1996). It was concluded that they were not
statistically different to the TOC ages they reported. They will be treated as TOC dates for the linear regression
analyses.

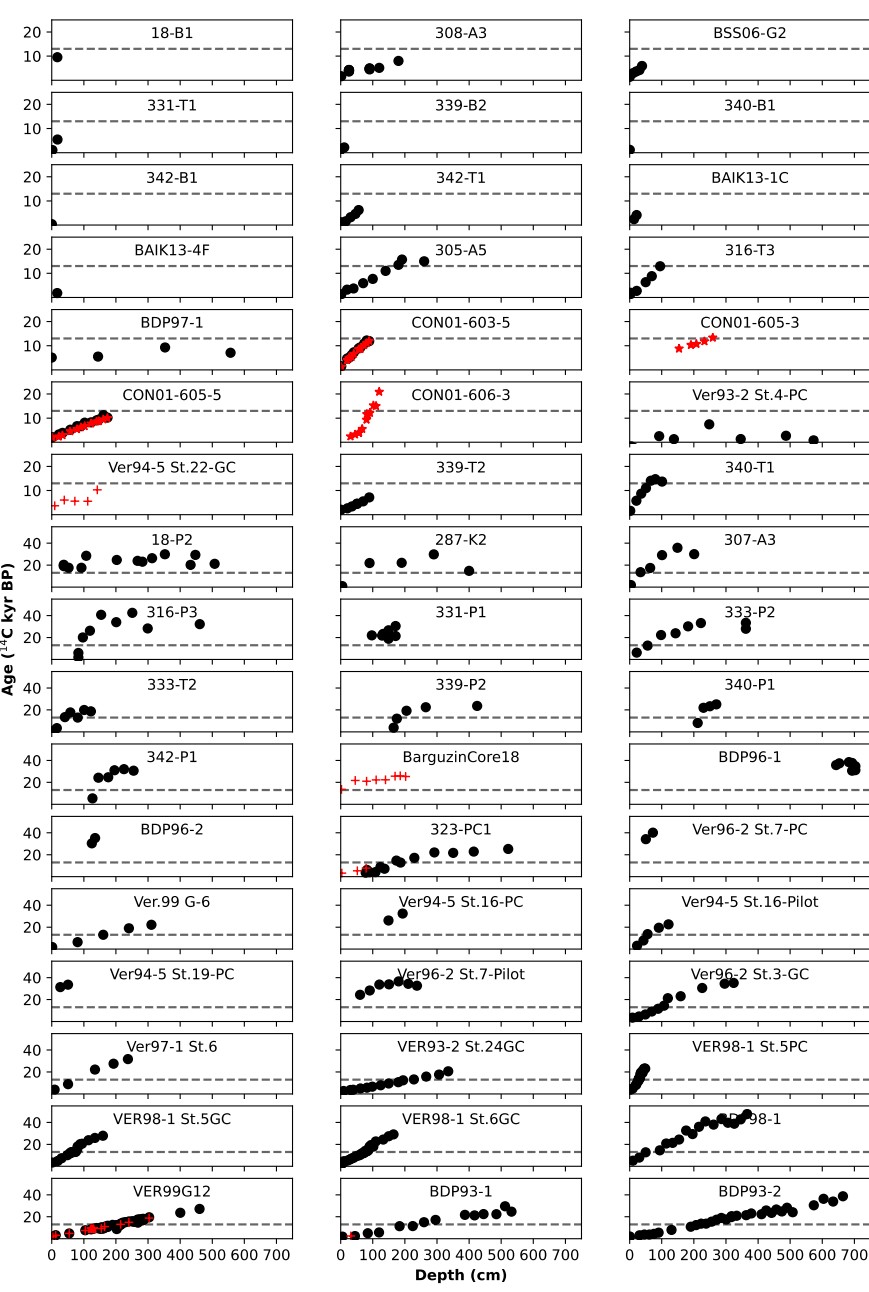

**Figure 4: Radiocarbon data from all 51 cores in this database, with mean uncalibrated radiocarbon age in $^{14}$C kyr BP on y axis and depth in cm plotted on x axis. TOC ages are shown as black dots, pollen ages as red stars, and all other materials (lipids, diatom/pelitic silt, wood) are shown as red crosses. The top two rows have smaller y axis limits to better show shorter cores. All x axes are the same. Vertical dotted lines are plotted at 13 $^{14}$C kyr BP to highlight the cut-off for our linear regression method.**



Several errors were found in Table 2 of Colman et al. (1996) providing depth values off by a factor of 10. These were cross-checked by contacting S. M. Colman and are reported correctly here. These errors were simply transcription errors, so no results are affected. Lab IDs and sample top/bottom depths for core BSS06 G-2 were added to this dataset by personal communication with Murakami. TOC radiocarbon ages for cores CON01-603-5 and CON01-605-5 are published here for the first time. Extra TOC ages for core VER98-1 St.6GC are also published here for the first time, supplementing the data published by Watanabe et al. (2009a). Finally, some lab codes that were wrongly transcribed in Nara et al. (2023) are corrected. Thirteen dates were reported with 'lower-bound' or negative radiocarbon ages ('>43240' or '-1300' respectively). We report these in a separate file for completeness but suggest not using them.

### 3.4 Age Offset Estimates from Linear Regression

Of the 51 cores with radiocarbon data reported in this compilation, 26 are used to calculate age offsets. In total, 19 estimates of apparent surface age (ASA) are made, using 142 TOC ages. To recap, the ASA is the y-intercept of the linear regression on TOC ages younger than 13 $^{14}$C kyr BP and represents an estimate of the age offset. The results for each core, grouped and summarised by their location are provided below.

### 3.4.1 Academician Ridge

| Core/Site | # of ages | ASA ($^{14}$C kyr BP) |
|---|---|---|
| Ver94-5 St.16 | 2 | -2.49* |
| 333 (2) | 4 | 0.08 |
| 331 (1) | 2 | 0.55 |
| 340 (2) | 5 | 1.28 |
| Ver96-2 St.3-GC | 5 | 1.94 |
| VER98-1 St.6GC | 16 | 2.17 |
| VER98-1 St.5 (1) | 9 | 2.54 |
| Ver97-1 St.6 | 2 | 2.77 |
| BDP98-1 | 3 | 2.86 |
| Mean | | 1.77 |
| Range | | 2.78 |

**Table 2: The ASAs ($^{14}$C kyr BP) for each core/site at Academician Ridge. Where cores were analysed as a composite, the number of cores from which data was used in the linear regression is denoted in parentheses. Cores with anomalous ASAs are marked with *.**

The ASA of 9 sites, using 11 cores, were returned from Academician Ridge (Table 2). Core Ver94-5 St.16 returned a negative age offset estimate however, hence we do not consider it in any further analysis, leaving 8 accepted ASAs. Cores 18-P2 and 18-B1 were left out as the former was non-linear and the latter only had one age. Core 340-P1 was left out because its only age younger than 13 $^{14}$C kyr BP was would have been a large reversal from the older ages of 340-T1 and was clearly erroneous. Core 307-A3 was left out because it only had one age younger than 13 $^{14}$C kyr BP. Cores 331-P1, Ver94-5 St.19-PC, Ver96-2 St.7-Pilot, Ver96-2 St.7PC, BDP96-1, and BDP96-2 were left out because they had no ages younger than 13 $^{14}$C kyr BP. Core Ver98-1 St.5PC seems to have suffered





from partial compression (clear from comparison to Ver98-1 St.5GC; Watanabe et al., 2009a) so was left out.

Lastly, core Ver94-5 St.22-GC didn't have any ages from TOC.

### 3.4.2 Buguldeika Saddle

| Core/Site | # of ages | ASA ($^{14}$C kyr BP) |
|---|---|---|
| 316 (1) | 6 | 0.92 |
| BDP93 (1) | 9 | 1.26 |
| 305-A5 | 6 | 1.34 |
| BSS06-G2 | 5 | 1.47 |
| 339 (2) | 8 | 1.48 |
| VER93-2 St.24GC | 11 | 1.75 |
| VER99G12 | 18 | 2.34 |
| Mean | | 1.51 |
| Range | | 1.42 |

**Table 3: The ASAs ($^{14}$C kyr BP) for each core/site at Buguldeika Saddle. Where cores were analysed as a composite, the number of cores from which data was used in the linear regression is denoted in parentheses.**

The ASA of 7 sites, using 8 cores, were returned from Buguldeika Saddle (Table 3). Core 339-P2 was left out due to its non-linearity (Figure 1), however, its inclusion did not impact our result if we reject the same ages Colman et al. (1996) rejects. Core 316-P3 was also left out due to its non-linearity. BDP93-1 was also left out, due to its suspected contamination by modern carbon (Colman et al., 1996). Including data from BDP93-1 would have changed the BDP93 ASA to 1.15 $^{14}$C kyr BP, similar to the estimate of 1.16 $^{14}$C kyr BP by Colman et al. (1996), yet still different due to the inclusion of ages published by Nakamura et al. (2003).

### 3.4.3 Other Locations

| Location | Core/Site | # of ages | ASA ($^{14}$C kyr BP) |
|---|---|---|---|
| Maloe More | 342 (3) | 7 | 0.50 |
| Posolskoe Bank | Ver.99 G-6 | 2 | 1.19 |
| Continent Ridge | CON01-603-5 | 10 | 1.89 |
| Vydrino Shoulder | CON01-605-5 | 12 | 1.62 |

**Table 4: The ASAs ($^{14}$C kyr BP) for each core/site in other regions. Where cores were analysed as a composite, the number of cores from which data was used in the linear regression is denoted in parentheses. No mean or range is provided here because they are each from different regions and so are not directly comparable as their own subset.**

The ASA of 1 site, using 3 cores, was returned from Maloe More. Continent Ridge, Vydrino Shoulder and Posolksoe Bank also had one estimate of ASA each, all from one core only. Core Ver.99 G-6 has a 10cm depth correction applied (Tani et al., 2002) after comparison with a corresponding multiple core M-6. Both CON01-603-5 and CON01-605-5 were suggested by Demske et al. (2005) to have had sediment missing from the core tops. Morley et al. (2005) find a depth correction for CON01-605-5 of 12.5cm based on correlation of pollen events, which we apply to this data but no such depth correction for CON01-603-5 has been provided, so it's ASA



may be an overestimate. All cores from the deep basins did not have ASAs. These deep basins are known for common turbidites which might significantly affect the age offset of the data and make for non-continuous sedimentation rates or disturbed sediment profiles, which would affect the method used here. Core BarguzinBay18 has no ages younger than 13 $^{14}$C kyr BP, so no ASA was calculated for that core (furthermore, the top 3cm of sediment returned a radiocarbon age > 13 $^{14}$C kyr BP, suggesting there has been erosion at this location, likely due to its shallow setting, rendering the site unsuitable for the linear regression method).

### 3.4.4 Overall

The mean ASA estimate for the whole lake, using our linear regression of ages younger than 13 $^{14}$C kyr BP, is 1.56 $^{14}$C kyr BP (Table 5). The median estimate is similar, at 1.48 $^{14}$C kyr BP. The minimum and maximum ASA estimates are 0.08 and 2.86 $^{14}$C kyr BP respectively, providing a very large range. The means for Buguldeika Saddle and Academician Ridge are similar to the mean of the entire lake, and their ranges overlap completely. The individual estimates from the other regions are all within the range of Academician Ridge (Table 5).

|  | No. of ASAs | No. of ages | Min | Max | Mean | Range |
|---|---|---|---|---|---|---|
| **Entire Lake** | 19 | 142 | 0.08 | 2.86 | 1.56 | 2.78 |
| **Buguldeika Saddle** | 7 | 83 | 0.92 | 2.34 | 1.51 | 1.42 |
| **Academician Ridge** | 8 | 46 | 0.08 | 2.86 | 1.77 | 2.78 |
| **Maloe More** | 1 | 7 | 0.5 | 0.5 | 0.5 | NA |
| **Continent Ridge** | 1 | 10 | 1.89 | 1.89 | 1.89 | NA |
| **Vydrino Shoulder** | 1 | 12 | 1.62 | 1.62 | 1.62 | NA |
| **Posolskoe Bank** | 1 | 2 | 1.19 | 1.19 | 1.19 | NA |

**Table 5: Summary statistics of all ASA ($^{14}$C kyr BP) estimates, when looking at different subsets, one of which being the entire lake.**

## 4 Discussion

### 4.1 Data Compilation

Whilst radiocarbon specific data compilation papers have been published for Lake Baikal before (Colman et al., 1996; Nakamura et al., 2003) this paper represents the first complete collection of all AMS data published before 2023 for Lake Baikal. Whilst most of the data we present is not of our own analysis, the paper represents a large step towards making all the data more accessible for future reuse. Having all data in one compilation, with transcription errors fixed, extra metadata, and some data made accessible for the first time will reduce the time needed to find/verify data of interest and may encourage those interested to utilise more data than they would have previously.



### 4.1.1 Poor Representation in Data Repositories

It was immediately apparent that archiving of radiocarbon data (and proxy data in general) from Lake Baikal into international data repositories is poor; compiling data using typical data repositories (Neotoma, Pangaea, NOAA) would have provided data from only three cores (searches done as of 1$^{st}$ July 2023): Neotoma contained 1 dataset for core CON01-603-5, but under a slightly different core name (CON16035); Pangaea contained datasets for CON01-603-5, CON01-605-5 and CON01-606-3, although data for core CON01-606-3 was reported twice with differing reporting standards; NOAA held no radiocarbon datasets from Lake Baikal. Incorrect core names – or inconsistent naming across publications, as was observed for a few cores in this study - means that simple data searches across multiple data repositories might report more data/cores than actually exist (i.e. CON16035 and CON01-603-5 might be reported as two different cores). Furthermore, interrogating the case of CON01-605-5 from Pangaea, this dataset is actually a composite core consisting of dates taken from neighbouring cores CON01-605-5 and CON01-605-3. While composite cores are certainly useful while presenting and analysing data for study, we only report datasets that are delineated by core (and we deconstruct composite cores into their original cores), as this helps highlight the origin of the data.

The lack of this representation in recognised data repositories means these data are not contributing to influential large scale data compilation or assimilation projects (Erb et al., 2022; Kaufman et al., 2020). Whilst reporting their radiocarbon data alone will not allow their inclusion in such studies, this study may act to spur proxy compilation work for Lake Baikal or the Baikal region.

### 4.1.2 Naming/Data Inconsistencies

The core naming inconsistency highlighted above was not unique. Multiple cores had different names across publications, which makes searching for them in the literature more challenging. Inconsistency in the spellings of different locations within the lake, such as five different spellings for Posolskoe Bank, may also make searching for relevant literature difficult. However, different spellings are to be expected across such a broad range of research, perhaps for cultural or linguistic reasons. We chose the more common spellings. There were also inconsistencies in the data reported for a single core between different papers. Some publications, when using data previously reported, seemed to reject ages without mentioning them whatsoever. This practice can be confusing for those interested in reusing the data and may lead to poor reuse.

### 4.1.3 Reporting Standards on Depth

What is starkly absent from all the reporting standards previously published are guidelines/recommendations on how to report the depth information regarding the position of the dated material from an archive. This is likely because different archives will have different ways of reporting position. This is vital information for constructing age-depth models (Heegaard et al., 2005), and even just in palaeolimnology there are a number of ways publications will report such data. These are: (1) reporting the top and bottom depth of the core sample; (2) reporting the middle depth and thickness of the core sample; (3) reporting just the middle depth of the sample. The last of these does not provide important thickness data. Our data format accommodates any of these reporting styles – the NOAA standard, which only accepts top and bottom depth data, not middle depth data, does not. The importance of the thickness data relates to what Heegaard et al. (2005) call the between-object error. Whilst many



age-depth models do not utilise such data, their ability to help better represent uncertainty means they should be
considered important data regardless. 56% of radiocarbon analyses in this compilation report thickness data.
Lacourse and Gajewski (2020) stress the importance of this metric after analysing a set of publications from 2018
and 2019 in *Quaternary Research* and *Journal of Quaternary Science*, finding that 75% of 34 papers they analysed
failed to report sample thickness.

### 4.2 Age Offset Estimates

The application of a single age offset estimation method to a number of cores within a single lake, or a single
region of a lake has been done before by Colman et al. (1996) (n=10 age offset estimates) and Watanabe et al.
(2009a) (n=3 age offset estimates) however this study represents the largest number of cores analysed with the
same method (n= 17 age offset estimates). The method used in this paper is similar to that of Colman et al. (1996).
The method of Watanabe et al. (2009a), by contrast, aligns positive anomalies in linear sedimentation rate to the
radiocarbon plateau of the Younger Dryas. We first discuss other results on the age offset for Lake Baikal, then
compare them to our own. The papers discussed below are not an exhaustive list of papers that utilise an age offset
estimate but focus on those that make some justification for their choice.

#### 4.2.1 Previous Age Offset Estimates

As previously mentioned, Colman et al. (1996) use a suite of linear regression methods to estimate the age offset
for cores in Lake Baikal, using either the topmost two ages in a core or all ages younger than 13 $^{14}$C kyr BP. The
cores they analyse come from either the Academician Ridge or Buguldeika Saddle regions (the latter called
Selenga Delta within their paper). They report that the age offsets from these two regions are distinct from each
other (0.47 ± 0.37 $^{14}$C kyr BP at Academician Ridge and 1.22 ± 0.18 $^{14}$C kyr BP at Buguldeika Saddle) and
hypothesise that the older age offset in Buguldeika Saddle may be due to an influx of older terrigenous sediment
from the Selenga River, with its outflow very near the Buguldeika Saddle but over 100km away from Academician
Ridge. Many papers use these results, however, most use the value of 1.16 $^{14}$C kyr BP, as returned from analysis
on BDP93, instead of using the summary statistics reported (Colman et al., 1999; Tarasov et al., 2007).

Karabanov et al. (2004), use a regression methodology to estimate an age offset of 1588 years from core VER93-
2 st.24GC, however do not describe whether all their dates are used for regression. This result was not reproducible
by us using any subset of their ages. It is possible this is because they report their ages as the 'analytical age'
which might suggest they are reported without the correction for isotopic fractionation however, correction for
isotopic fractionation is mentioned in their methodology and δ$^{13}$C data is provided. Many of their dates are reused
by Tarasov et al. (2007) in the same form reported by Karabanov et al. (2004). We therefore interpret this to mean
that the data reported by Karabanov et al. (2004) do not need further isotopic correction.

Demske et al. (2005) estimate the age offset of pollen concentrate ages (not the TOC age offset) by performing
three linear regressions, however the number of ages used for each regression is not described. For core CON01-
603-5 (Continent Ridge) they use the top three ages to get a value of 930 $^{14}$C yr. For core CON01-606-3 (Posolskoe
Bank) they report a value of 675 $^{14}$C yr and for the composite core consisting of cores CON01-605-3 and CON01-
605-5 (Vydrino Shoulder) they report an apparent surface age of 960 $^{14}$C yr, however we could not reproduce



either of these using any data combination of theirs in a regression. Note these results are from pollen concentrates,
which likely have a different age offset to TOC. The non-zero nature of these offsets however highlights that
pollen concentrates still suffer from an age offset.

Prokopenko et al. (2007) argue that a Lake Baikal TOC age correction "should not exceed 500yr", and argue
against the previously proposed age offset estimates of greater than 1000 $^{14}$C yr. Their reasoning represents the
most involved discussion of the age offset for Lake Baikal, so we take time here to examine their different claims.
Their reasoning begins with a suggestion that linear regression-based approaches to estimating the TOC age offset
have produced contradicting results for nearby cores, such as the 1160 $^{14}$C yr estimate for BDP-93 (Colman et al.,
1996) and the 1588 $^{14}$C yr estimate for VER93-2 St.24GC (Karabanov et al., 2004), both from the Buguldeika
Saddle. They use the argument that a "true reservoir effect for a lake cannot be core- or site-specific" to refute
these two estimates and their methodologies. Tangentially, we highlight that reservoir effects could indeed be
site-specific, as they are in the ocean, highlighting that Lake Baikal is the largest lake by volume in the world with
three large basins. Although, when correcting radiocarbon ages before calibration, it is their age offset, not solely
their reservoir effect, that must be accounted for, and this can clearly have site-specific component in theory, for
example if the regions receive different influx, as the Buguldeika Saddle and Academician Ridge areas do. Noting
that the two cores highlighted above are both in the same region, we agree that they should both have the same
age offset, however instead of disregarding the differing results of Colman et al. (1996) and Karabanov et al.
(2004) as problematic, like Prokopenko et al. (2007), we argue that they can still inform us of the true age offset,
in the same way as any uncertain estimator can. The key is simply to recognise that there is uncertainty involved.
Prokopenko et al. (2007) also suggest a "critical cross-check" for the TOC age offset is available in the
radiocarbon ages of the twin BDP-93 cores, referencing a wood age and a TOC age that are from similar depths
in each core. The wood age is approximately 500 years younger than the slightly deeper TOC age, so imposing
an offset of over 500y on the TOC age creates a stratigraphic reversal, the deeper age now being younger. This
supposed contradiction, however, doesn't account for the fact that wood ages are also known to have age offsets
(Hatté and Jull, 2013). Indeed, Oswald et al. (2005) compare the ages of different macrofossil types in Arctic
lakes and find that "wood and charcoal are generally older than other macrofossils of the same sample depth with
age differences ranging from tens to thousands of years". They attribute this to the decay-resistance and/or the in-
built age of woody macrofossils. Similarly, Prokopenko et al. (2007) utilise the radiocarbon ages from pollen
extracts in core CON01-603-5 to interpolate an age of ~3,500 $^{14}$C yr BP for a lamina enriched in the diatom
*Synedra acus*. Comparing the interpolated pollen date of this lamina in CON01-603-5 to the TOC ages of similar
lamina in cores 305-A5, BDP93-2 and Ver93-2 St.24GC (3,710 ± 60; 3,800 ± 35 and 3,800 ± 45 $^{14}$C yr BP) they
suggest the difference in radiocarbon age of only ~300 $^{14}$C yr is consistent with a 500-yr adjustment to bulk TOC
ages. Again, this doesn't account for the fact that pollen concentrate ages can exhibit age offsets (Kilian et al.,
2002; Neulieb et al., 2013; Schiller et al., 2021), through for example, contamination and redeposition. These two
instances of mistaking dates of terrestrial material as being free of age-offsets highlight here the utility in using
the term age-offset, instead of reservoir age. The fact that terrestrial material is free of a reservoir age does not
mean it is free of an age offset. The last line of reasoning invoked for a ~500 $^{14}$C yr offset is that the average of
the youngest three dates returned from Lake Baikal is 545±145 $^{14}$C yr. Two of these dates are TOC from Maloe
More (and reported in our dataset) and one is from a modern sediment trap that collected diatomaceous sediment





(Colman et al., 1996). However, ages from diatomaceous sediment cannot be used to infer the age offset for TOC.
Without this, the remaining two dates from Maloe More, which differ by over 400 $^{14}$C yr, are not suitable to infer
an age-offset for the rest of the lake.

Watanabe et al. (2009a) present radiocarbon dates from three cores in Academician Ridge (two are from the same
site), each showing a region of paired positive and negative linear sedimentation rate (LSR) anomalies. These
events all show anomalously low apparent sedimentation rate and then anomalously high apparent sedimentation
rate before returning to 'normal' sedimentation rates at 12.1 $^{14}$C kyr BP or 12.2 kyr BP. Several explanations for
these LSR anomalies are ruled out before settling on the possibility that they represent the radiocarbon plateau of
the Younger Dryas (YD). Using a calendar age of 11.6 cal kyr BP for the end of the YD, they de-calibrate this to
10.1 $^{14}$C ka BP and calculate a 2100 ± 90 $^{14}$C yr correction to match their LSR anomaly dates to the end of the
YD. The uncertainty of their estimate does not include the uncertainty of the de-calibration, however.

Nara et al. (2010) apply an age offset of 500 yr to TOC dates, following Prokopenko et al. (2007) and also to
pollen concentrate dates from core VER99G12. The latter is justified by noting that Boës et al. (2005) found a lag
of ~500 yr between the GISP2 $\delta^{18}$O record and a record of grayscale fluctuation from core CON01-603-5 attached
to the pollen chronology of Demske et al. (2005). Recognising offset predicted by Watanabe et al. (2009a) of
2100 +/- 90 $^{14}$C yr at Academician Ridge, they suggest that this lower offset at Buguldeika Saddle may be due to
a large input of modern organic material from the Selenga River. Coincidentally, this is the mirror image of the
reasoning Colman et al. (1996) used to explain the higher offset they found at Buguldeika Saddle compared to
Academician Ridge, suggesting the Selenga may have provided older carbon material.

Murakami et al. (2012) use an age offset value of 1418 $^{14}$C yr. This is inferred from a radiocarbon date from depth
0-1cm in their core BSS06-G2, reported with an age 1418±36 $^{14}$C yr BP, assuming that this sediment should be
approximately modern. No uncertainty is included here.

Nara et al. (2023) correct for a reservoir effect of 380 $^{14}$C yr in core VER99G12, due to the 380-yr water residence
time of the lake measured by Shimaraev et al. (1993). There is no reason the residence time of water should impact
the reservoir age, however.
**4.2.9 Our Age Offset Results**
We return 19 age offset estimates from cores across the whole lake. The minimum (accepted) estimate was 0.08
$^{14}$C kyr BP, and the maximum estimate was 2.86 $^{14}$C kyr BP. The range of these estimates is greater than the range
of estimates in the previous literature. Both the minimum and maximum estimates came from the Academician
Ridge, which had 8 estimates of age offset: The range from Buguldeika Saddle, which had 7 estimates of age
offset is significantly lower, with a regional minimum of 0.92 $^{14}$C kyr BP and a regional maximum of 2.34 $^{14}$C
kyr BP. The lower range in Buguldeika Saddle is likely related to the higher sedimentation rate, which typically
leads to higher quality radiocarbon dates as it reduces the effect of bioturbation, which cannot be ruled out in Lake
Baikal (Martin et al., 2005).



The mean and standard deviation of the estimates from each site are 1.77±1.04 [14]C kyr BP for Academician Ridge
and 1.51±0.45 [14]C kyr BP for Buguldeika Saddle. To test whether we can argue the Academician Ridge or
Buguldeika Saddle have different age offsets we use a Welch's T-Test. This returns a p-value of 0.527, so we
cannot reject the null-hypothesis that these regions have identical age offsets. Furthermore, the estimates from the
other 4 regions of the lake are all within the range of estimates from Academician Ridge, and hence we cannot
argue that the age offset of the lake differs between different regions of the lake. Given the uncertainty around
measuring age offsets, it is not surprising that we cannot statistically argue this. The result has interesting
implications for considering the source of the age offset, however. A region-specific age offset may be explained
by some source of old carbon entering the system and having a local effect, for example through the Selenga River
as was proposed by Colman et al. (1996). However, considering the Academician Ridge is approximately 200km
from the Selenga Delta, it is unlikely that particles of old sediment would influence Buguldeika Saddle and
Academician Ridge in a similar manner. Uncovering the source of this age offset still poses a challenge.

Given that we cannot argue that there is a systematic age offset impacted by lake location, we suggest that the
best estimate of age offset in the lake will be the mean and standard deviation of all estimates throughout the lake,
which is 1.56 ± 0.75 [14]C kyr BP. A more precise estimate might be reasoned for using only the data from
Buguldeika Saddle, which provides an estimate of 1.51± 0.45 [14]C kyr BP. This might be justified by reasoning
that Buguldeika Saddle, with its high sedimentation rate, provides the best estimates. However, we believe that
more data is needed from that region of the lake to solidify those results. Further, we do consider whether the
large uncertainty in our estimate is due to the use of sediments that are geographically distant, for even within our
regional groupings of Academician Ridge and Buguldeika Saddle, some cores are over 10km apart. To this, we
highlight a cluster of cores within the Buguldeika Saddle area that are very close together, and can, with high
confidence, be expected to have experienced the same sediment input: cores BDP93, 339, VER93-2 St.24GC, and
VER99G12. These 4 core sites each provided an ASA, and are within 2km of each other, and yet the ASA
estimates are 1.26, 1.48, 1.75 and 2.34 [14]C kyr BP respectively. This lends weight to our reasoning that the
different ASAs are not due to environmental differences. but are due to errors inherent to estimating the age offset
with this method. We argue, therefore, that any estimate of age offset should, for Lake Baikal, incorporate
uncertainty of at least 0.5 [14]C kyr BP. We suspect that the major cause of our uncertainty is error in the radiocarbon
dates themselves, although the variable loss of sediment from retrieved cores and error related to assumption of
constant sedimentation rates and constant age offsets are likely to contribute to the uncertainty too.

Lastly, we have highlighted that the method of using a linear regression to estimate the age offset has uncertainties
of hundreds of years. Linear regression is likely to provide a more accurate answer where sedimentation rates are
high, but it should not be used where turbidites or variable sedimentation break the assumption of constant
sedimentation that is required for the technique. Ideally, multiple cores should be taken/used when using this
technique to estimate age offset. This should not discourage investigators from using a linear regression method
to estimate age offsets (when appropriate), however. Other methods, such as taking a surface sample or comparing
to some perceived known date, may seem to have lower uncertainty, however this uncertainty is likely less well
constrained and may be just as large. A further implication of our result is that many previous studies are likely
to have significantly underestimated the uncertainty in their estimates of age offset. The promise of reliable



radiocarbon dating free of age offsets through a new technique preparing pollen concentrates by Omori et al.
(2023) is particularly exciting in light of our results.

### 5 Data Availability

The data can be accessed at https://doi.org/10.1594/PANGAEA.973799 (Newall et al., 2025).

### 6 Conclusions

In this study, we have (i) created a complete database of all AMS radiocarbon dates from Lake Baikal published
before 2023, standardising the reporting, updating missing or incorrect metadata, and adding some previously
unpublished dates, (ii) produced a new estimate of age offset for TOC in Lake Baikal sediments of $1.56 \pm 0.75$
$^{14}$C kyr BP, and (iii) did not find evidence to suggest that different regions of Lake Baikal have a statistically
different age offset, as previous studies have suggested. The primary implication of our results is that previous
Lake Baikal studies have significantly underestimated the temporal uncertainty from radiocarbon results. More
generally, however, our study has shown that a linear regression method for estimating age offsets has a large
inherent uncertainty that has likely been underestimated when used in other lakes/previous studies. Other
techniques for estimating age offset should be examined in a similar manner to evaluate their uncertainties: It is
not clear that other methods would have less uncertainty than a linear regression method. We hope that this study
facilitates further research in Lake Baikal by improving access and understanding of previous radiocarbon work
that has taken place, and spurs on further work to understand the uncertainties in estimating radiocarbon age
offsets.

### 7 Author Contribution

Conceptualisation: SN and AM
Data Curation: SN
Formal Analysis: SN
Investigation: SN and NP
Methodology: SN
Project Administration: SN and AM
Software: SN
Supervision: AM
Visualisation: SN
Writing: original draft preparation: SN
Writing: Review and Editing: SN, AM and MB

### 8 Competing Interests

The authors declare that they have no conflict of interest.



## 9 Acknowledgements

Many thanks to Miles Irving for creating the map figures.

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
