# Peer review of "Insights into Lake Baikal Radiocarbon Age Offsets from a"

_Earth System Science Data, 2025_

## Author Comment (AC1)

Dear Editor & Reviewers,

We are grateful for your patience with our revisions and to both you and the reviewers for helping this manuscript find its way to publication.

Below we detail our responses to the two reviewers. Their comments stimulated discussion and analysis that improved the rigor and clarity of the manuscript immensely. The major changes, in summary, are:

- Description of methods for any dates published for the first time in place of reference to "(Piotrowska, personal communications)".

- Addition of a summary figure (Figure 5) presenting the spread of ASA results from different regions of the lake.

- Significant rewriting to make the tone more suitable for a journal.

- Mention of our analysis on TOC pretreatments to address query by Reviewer 1.

- Provision of analysis code in an interactive computing environment on Zenodo to allow others to replicate and evaluate our code and any subjective decisions regarding removal of outliers.

Line numbers refer to the line number in the revised manuscript (after tracked changes accepted) unless otherwise stated. References cited in our responses are provided at the bottom of the document.

Best,

Sam Newall

**Reviewer 1 Comments**

1. *Lake Baikal is one of the most extensively studied lakes in the world. The compilation of published radiocarbon ages from numerous lake sediment cores represents a valuable Earth System Science dataset, and this article provides important context to support it. The dataset will serve future researchers interested in a variety of environmental changes recorded by Lake Baikal sediments. Although this audience may be relatively small, the authors could broaden the significance and novelty of their work by offering a model for reporting radiocarbon ages from sedimentary "total organic carbon" (TOC).*

Authors' Response: We thank the reviewer for their useful comments and complimentary reviews of the manuscript. We have added commentary about the need to better adhere to the standards already suggested for reporting radiocarbon ages (Stuiver & Polach, 1977; Millard, 2014) but are hesitant to formally suggest a model or standard for reporting TOC ages as the precedent for suggesting such standards is to collect ideas/opinions/thoughts from the wider community (i.e. Khider et al., 2019; Millard, 2014; Stuiver and Polach, 1977), which we did not do. So, whilst we think the suggested work would be valuable, we do not believe this work is suited to provide that.

2. *There are different "flavors" of TOC (also called, bulk sediment) depending on how the sediment is pretreated prior to 14C measurement. The choice of procedure can significantly influence the resulting 14C age. Documenting which pretreatment method was used for each sample is necessary not only to compare results across laboratories, as is done in this article, but also to better understand the sources of old carbon underlying the age offsets and thereby to address the authors' motivation for compiling the dataset– for "insights into radiocarbon age offsets" as stated in the article's title.*

Authors' Response: We thank the reviewer for suggesting we critically review sediment pretreatments prior to 14C measurement. We reviewed every Baikal study where sediment samples had been analysed for radiocarbon dating, paying special attention to the methods used in the radiocarbon analyses, especially the pretreatments. These are summarised in the Table below, but specifically we looked at every method for information on:

- Whether sediment samples were sieved to remove macrofossils or ground to incorporate particulate organic matter, which some scientists consider part of TOC

- Whether sediment samples were acidified to remove inorganic carbon (decarbonated)

- Whether sediment samples were filtered, and at what pore size (typically 1 μm), which removes fine clay and other particles along with adsorbed organic carbon.

- Whether sediment samples were treated with an alkaline solution to remove base-soluble organic carbon (humic acids), leaving behind humins, which are nearly always older than the total 14C population of the sediment

We note that wood samples in (Watanabe et al., 2007, 2009b) were treated with 1.2 M HCl and 1.2 M NaOH, but these samples are not considered in the table below as our study is focussed only on sedimentary TOC

All pretreatments with the exception of Fedotov et al. (2024) report an acidifying stage

Only Colman et al. 1996 undertook sieving of the sedimentary material before analysis to remove macrofossils. They found that macrofossils are extremely rare in Baikal deep-water sediments, and although they did experiment with separating out organic matter < 63 um and then dating fractions <63 um and > 63 um in the same samples, they found that there was "no consistent relationship between the age of these fractions and the corresponding TOC ages" and instead focussed on only dating TOC. These findings likely contribute as to why no sieving has been undertaken in other studies.

Only Colman et al. 1996 specified that they filtered acidified TOC samples. No studies on sediment samples were treated with an alkaline solution to remove humic acids.

Therefore, we believe all the ages we used in our analysis underwent comparable pretreatment. Text has been added to state this belief but highlight the possibility of differences in pretreatment on Line 520-523.

Table R1: Pretreatment steps mentioned in papers describing TOC radiocarbon ages. N=no or not mentioned; Y=yes

| Study | Sieving | Acidified | Filtered | Alkaline soln |
|---|---|---|---|---|
| Colman 1996 | Y | Y | Y (1um) | N |
| Tani 2002 | N | Y | N | N |
| Nakamura 2003 | X | Y | N | N |
| Karabanov 2004 | X | Y | N | N |
| Watanabe 2007 | X | Y | N | N |
| Watanabe 2009a QI | X | Y | N | N |
| Watanabe 2009bEPSL | X | Y | N | N |
| Nara 2010 | N (except pollen) | Y | N | N |
| Murakami 2012 | N | Y | N | N |
| Swann 2020 | N | Y | N | N |
| Nara 2023 | N | Y | N | N |
| Fedotov 2024 | N | N | N | N |

3. *The article would be strengthened by extending the context beyond age modeling to recognize that age offsets have been extensively studied in the broader context of watershed carbon cycling in both lake and marine depositional settings. This literature includes reports of downcore changes in age offsets, which reflect shifts in carbon cycling and directly affect the use of downcore trends to estimate long-term TOC age offsets and associated uncertainties, the focus of the data analysis in this article.*

Authors' Response: We agree that this work would be of use and interest to those who use age offset results to evaluate changes in watershed carbon cycling. In particular, the evidence of large (but hidden) uncertainty around a given estimate of age offset should suggest caution when interpreting different estimates of age offset throughout a core as truly representative of downcore changes in age offset. However, because our method does not allow for downcore analysis of age offset, and because Lake Baikal age offsets have not been used in this way in the literature, it is outside the scope of the work we have done for this project, so we leave it to the interpretation of the reader.

4. *For results from different laboratories to be compared with confidence, the specific fraction of organic carbon analyzed should be the same. At first glance, the offset ages appear higher for samples measured in the Nagoya lab than in those from Woods Hole. If so, this difference may reflect pretreatment methods: the former retains all organic carbon (in-capsule acidification), while the latter removes acid-soluble organic carbon as well as organic carbon adsorbed to fine particles (acid washing through a 1 µm filter). While I am not aware of any published studies that quantify the difference in the outcome of these two procedures for 14C, I hypothesize that acid washing and filtration preferentially remove a younger fraction of 14C. I suggest that the authors use their dataset to test the possibility that the age offset depends on how the sediment is pretreated. This could contribute to the goal of using the dataset for "insights into radiocarbon age offsets."*

Authors' Response: As from our response to point #2 above, all ages that were used for age offset estimation in this paper performed some acidification or decalcification step. We did not evaluate in further depth the differences between the acidification processes from Nagoya and Woods Hole, assuming that differences that were not mentioned in the literature were not going to be substantial. We had not noticed that the ASAs from the Woods Hole dated cores tended to be lower than those for the Nagoya dated cores. We would like to highlight, however, that most Woods Hole dated cores are from the Colman et al. (1996) paper, so it is possible that some other common factor came into play. Furthermore, the only Woods Hole dated core not from Colman et al. (1996) is Ver93-2 St.24GC (Karabanov et al. 2004) and this provides one of the higher offsets in the Buguldeika Saddle region. Given that BDP93-1, one of the cores studied in Colman et al. (1996), suffered from significant modern carbon contamination due to poor storage, it is possible that modern carbon contamination could also be responsible for the difference between the Colman et al. (1996) cores and the Japanese studied cores. Due to these possible confounding factors, we choose not to make any inference about the difference between the procedures from our data. We'd prefer to leave that to a dedicated experiment comparing the two procedures.

5. *Furthermore, the surface sediment ages reported by Fedotov et al. (2023) are conspicuously old. Although I could not easily determine the protocols used by the Golden Valley AMS lab, I wonder whether the sediments were treated with an alkaline solution to isolate the residual humin fraction, which could explain the anomalously old surface ages. I also note that the dataset lists the "dated material" for these samples as "bulk silty clay" rather than "TOC," which highlights the need for a more precise documentation of what was actually analyzed.*

Authors' Response: Unfortunately, we did not receive a response from Fedotov about their pre-analysis procedure. However, we do offer some comment here on why their surface ages are conspicuously old. The core was extracted from relatively very shallow water (41m) in the Barguzin Bay, opposite the inflowing rivers of the Barguzin River, Baikal's second largest inflow. The authors themselves conclude that organic late glacial / Holocene sediments have been removed prior to core extraction, most likely by underwater current erosion from the Barguzin inflow. That very few diatoms were present, and BiSiO2 was very low confirms diatomaceous sediments were not collected. In conclusion, we do not expect that these dates are conspicuously old because of the way the samples were treated, but because the upper sediments had already been removed prior to core extraction.

6. *A number of radiocarbon ages in the dataset are cited as "personal communication." The associated pretreatment procedures are therefore not available. Moreover, including unpublished data seems inconsistent with the authors' statements that the dataset is derived from published sources. If this article is the first publication to report these ages, then say so; if not, cite the publication.*

Authors' Response: There are three distinct groups of data for which the term "personal communication" was used in the dataset. The first group are the TOC ages from cores CON01-605-5 and CON01-603-5, which had references of "Piotrowska (personal communication)" - these will be changed to "This study". These are dates analysed many years ago by one of the authors (N. Piotrowska) that had previously not been reported/mentioned in any previous paper. We

have added a section to the Methods (section 2.2) to highlight this and describe the pretreatment method.

The second group contains ages that were mentioned or plotted in previous papers but for which the radiocarbon data were not formally provided in the paper. These had references of only "Watanabe (personal communication)" and were from cores VER98-1_St.6GC and VER98-1 St.5PC. These dates are plotted in Fig. 2 of (Watanabe et al., 2009a) but their data are not provided in the data table, Table 3, of the paper. We acquired the data from email correspondence with Takahiro Watanabe. Because the data was not provided in a paper, we decided to simply reference them with "Watanabe (personal communication)". We have decided to change the references to "Watanabe et al. (2009a)", as the data were mentioned (albeit not provided) in this paper.

The third group contain ages for which the basic data were provided in some paper but extra information were obtained from communication with the author. These have references of a paper and of a personal communication, for example "Swann et al. (2020); Swann (personal communication)" or "Watanabe et al. (2009a); Watanabe (personal communication)". On review, not all dates for which extra information were provided have been treated in this way. These will be changed to remove personal communications from the reference column.

7. *"A full accounting of what constitutes "TOC" should include:*
    1. *Whether sediments were sieved to remove macrofossils or ground to incorporate particulate organic matter, which some scientists consider part of TOC.*
    2. *How sediments were acidified to remove inorganic carbon (decarbonated), either:*

        *a. in-capsule, which retains acid-soluble organic carbon (fumigation or liquid), or*

        *b. by rinsing, which removes acid-soluble organic carbon.*
    3. *Whether the sediments were filtered, and at what pore size (typically 1 μm), which removes fine clay and other particles along with adsorbed organic carbon.*
    4. *Whether the sediments were treated with an alkaline solution to remove base-soluble organic carbon (humic acids), leaving behind humins, which are nearly always older than the total 14C population of the sediment.*

Authors' Response: We refer to our response to point #2, and the table provided there.

8. *A dataset designed to provide "insights into radiocarbon age offsets" would also benefit from including, where available, measurements of both carbon and nitrogen abundance in the analyzed material. Alongside δ13C, which is already included, these data can help estimate the proportion of terrestrial versus aquatic carbon sources. Moreover, carbon concentration (C%) is typically inversely related to offset age and can be used to estimate how age offsets vary downcore.*

Authors' Response: We did provide carbon concentration (previously called carbon yield, we have changed to referring to it as carbon content). We did not include nitrogen measurements because the vast majority of the dates did not have any corresponding nitrogen measurements. We agree that such auxiliary information might be useful in better understanding age offsets, however we could not evaluate that in any way with this dataset.

9. *In my view, the text requires extensive revision to adopt a style more appropriate for a journal audience. As it stands, much of the article reads like a first draft. It could be shortened by at least one quarter, perhaps one third, by eliminating repetition, trimming unnecessary words and phrases, using more precise wording, and moving details of specific corrected errors into the dataset's "comment" field.*

Authors' Response: We have revised the style somewhat to make it more concise. Examples of where text has been shortened or removed include: Abstract (lines 11-31) has been made less descriptive; Paragraph describing Lake Baikal characteristics (Line 101 to 112 in initial submission) has been removed; Removed mention of data repositories

and reduced conversational language in Data Organisation section; Removed conversational language in section 3.1 Core Data Overview; Paragraph discussing Prokopenko et al (2007) results starting line 445 has been significantly shortened.

> 10. Finally, the article would benefit from a concluding figure that clearly illustrates its main findings. Figure 4 provides a useful overview of all ages, but a new figure focusing on the 19 accepted core chronologies is needed. I suggest plotting ages by depth for each of the six regions in Table 5, using distinct symbols for different cores from Buguldeika Saddle and Academician Ridge.

Authors' Response: We thank the reviewer for suggesting a concluding figure. We have added Figure 5. We decided to focus on a visualisation of the ASA results, focusing on the results and conclusions, instead of the accepted core chronologies as suggested. In recognition of the need for some visualization of the accepted chronologies we have provided access to the Jupyter Notebook that documents our analysis, code, and provides figures equivalent to Figure 1 for each accepted site (see Section 6 Interactive Coding Environment, starting Line 567)

> 11. In summary, the manuscript presents a valuable compilation of radiocarbon ages from Lake Baikal sediments. However, the manuscript requires substantial revision before it I would recommend it for publication. In particular, greater attention is needed to (1) clearly documenting what constitutes "TOC" across laboratories by specifying the pretreatment method used for each sample, (2) engaging more fully with the literature on age offsets, including those that specifically investigate temporal changes in age offsets, and (3) improving clarity and conciseness throughout the text.

Authors' Response: We thank for the reviewer for allowing us the opportunity to make substantial revisions, especially regarding what constitutes TOC and specifying pretreatment methodologies and improving clarity and conciseness throughout the text.

> 12. Line 11:The article does not address "contamination." I recommend avoiding confusion between age offset, which generally reflects TOC ages older than the time of deposition, and contamination, which results an age that is too young.

Authors' Response: Contamination is now mentioned in the article (Line 61, Line 101). See our response to the comment about line 56 which also discusses contamination and whether it should be intertwined with age offset.

> 13. Line 15 and throughout:When presenting age offset values, omit "BP." Offsets are reported as (radiocarbon) years, not years "before present."

Authors' Response: Age offsets are now reported as 14C kyr. Apparent surface ages (ASAs) are still reported with 14C kyr BP.

> 14. Line 48 and elsewhere:"Age-offset" does not require a hyphen.

Authors' Response: We have made sure we stick consistently with "age offset".

> 15. Line 36:Not all radiocarbon ages require offset correction.

Authors' Response: Whilst we agree that not all radiocarbon ages require offset correction, we believe that for all ages there should be a careful consideration of whether or not age offset correction is required, hence it is always a part of the process. We have changed the sentence to highlight that it is not always necessary, in the following way: "The process of using radiocarbon dates includes age offset correction (if applicable), calibration, and age-depth modelling…" (Line 40).

> 16. Line 49:Replace "true age" with "timing of deposition" or similar phrasing.

Authors' Response: Have replaced "true age" with "depositional age" (Line 52).

17. *Line 56:The hardwater effect involves more than "the presence of carbonate rocks in the lake" and can significantly contribute to age offsets. It deserves more attention in the article, even if it is not a significant issue in Lake Baikal. Also, I recommend not including contamination as part of the definition of age offset, since contamination is typically spurious, post-depositional, and unrelated to watershed carbon cycling—unlike offsets, which provide meaningful insights into the carbon cycle.*

Authors' Response: Our understanding is that the hardwater effect is linked to ancient carbon that has been dissolved into the water from bedrock, with a link to how water becomes "hard" if it has lots of dissolved ions. We have changed our description of it to be more encompassing: "the presence of carbonate bedrock within the watershed which supplies the water with old, radiocarbon-free dissolved inorganic carbon (DIC), known as the hardwater effect (Philippsen, 2013). " (Line 60). We understand the desire for it to have more attention in the article however we feel that it is too far outside the scope of the article. Firstly, because it is not typically considered a significant issue in Lake Baikal. Secondly, because it belongs in a discussion about the causation of the age offset whereas this paper primarily addresses quantification of the age offset. Thirdly, because we found definitions of causative effects, such as reservoir effect and hardwater effect, to be inconsistent and confusing, making discussing and evaluating those effects difficult.

Regarding contamination, we have added a few sentences to the introduction to better convey what we consider to be contamination: "Another potential contributor to the age offset, which we consider to be different to the reservoir effect, is contamination by both young and old organic material, due to: deposition and reworking of older sediments (known as the old carbon effect); bioturbation; root penetration; and infiltration of humic acids may also contribute towards age offsets (Björck and Wohlfarth, 2002). Contamination that occurs post-coring, such as in core storage or transport, we do not consider a contributor to age offsets." (Line 61). We think some contamination (pre-coring, i.e. old carbon effect) should be considered related to age offsets whilst some contamination (post-coring) should be considered a source of error.

18. *Line 75:Change "2009" to "2009a" or "2009b."*

Authors' Response: Have changed "2009" to "2009a". Thank you for catching this.

19. *Line 79:Replace "Ramsey" with "Bronk Ramsey" here and in the references.*

Authors' Response: Have changed "Ramsey" to "Bronk Ramsey" here and in references. Thank you for catching this!

20. *Line 137:When reporting depth, specify the datum: (1) core interval top (often embedded in the sample ID), (2) lake bottom (below lake floor), or (3) lake water surface.*

Authors' Response: We do not recognise these datums from any of the literature from which we collected depth information. The data is all reported as was reported in the original papers, for which we believe the (unspoken) convention is that the depth is reported with core interval top as datum. It seems that any corrected depths were applying corrections to change the datum to the lake bottom. We have added clarification to the paragraph regarding depth as follows: "We provide sample depth as a combination of the top, middle, bottom depth and thickness of the sample based on how the information was presented in the original paper or in our communication with the original author. All these depths are presented with the core top as the datum. Where cores had depth corrections for estimated loss of sediment at the top of the core (e.g. Colman et al., 1996; Morley et al., 2005) we provide a corrected middle depth for each sample. Corrected depths have the lake bottom as their datum." (Line 150)

21. *Line 140:If the template is not publicly available, there is no need to advertise it.*

Authors' Response: We have removed language about the "template".

22. *Line 141:The claim that the dataset "draws heavily from Millar (2014) and Stuiver and Polach (1977)" is confusing. Millar (2014) identifies six elements that should accompany each radiocarbon age, three of which concern calibration, which is not addressed in this article, and one of which is "pretreatment*

*procedure,"* which I strongly recommend including. Stuiver and Polach (1977) focus on conventions for radiocarbon age calculations, which do not appear to be central to the reporting template

Authors' Response: We have rewritten the portion regarding the reporting template and made it more clear which parts are based on each of these two papers. (Line 142-146) The Stuiver and Polach (1977) reference is the original paper setting up the convention to report dates as "conventional" dates and to use "$1\sigma$" error (a convention that has not since changed) and defines "conventional" dates. All our data adheres to this standard. The three elements concerning calibration are not of relevance to this paper as we perform no calibration.

Originally, we had planned to have "pretreatment procedure" as part of the dataset but it was cut out because a standardized reporting approach for it could not be found. We have added a section in the paper to describe the results of our evaluation of pretreatments in wake of your comments. (Line 520).

23. *Line 147:Provide details on how the reported depths were "corrected."*

Authors' Response: Details on how depths were "corrected" are available in the metadata .txt file within the dataset in Pangaea. We thank the reviewer for highlighting that this information was not made clearly accessible. For clarity, all depth corrections reported/used in this dataset/paper are following corrections made by the original authors. Core depth correction is common in studies where multiple cores from the same location have been investigated (Colman et al., 1996; Morley et al., 2005). A variety of techniques are used. For example, Morley et al. (2005) transferred a radiocarbon chronology carried out on a box core from the Vydrino Shoulder in the south basin of Lake Baikal to a piston core taken from the same location at the same time. The two cores were matched together using specific peaks in certain diatom species over the Holocene period. However, further diatom analyses of a gravity core taken from the same location at the same time (that was known to have preserved the surface sediment - water interface) revealed that both the box and piston core were missing their upper-most sediments so a new age-depth profile was developed taking this missing sediment into account.

24. *Line 149:Among the δ13C methods listed in the dataset is "AMS." My understanding is that graphitization and ionization during AMS analysis can cause δ13C fractionation. While measured and used to calculate 14C ages, the AMS-derived δ13C may not reflect the true value. For clarity, replace "AMS" with "Approximated from AMS" or include a note of explanation.*

Authors' Response: Yes, this is our understanding too. To clarify, we have included the following note of explanation "AMS-derived δ13C values may have undergone fractionation during the AMS process hence may not be representative of the true sample value." (Line 144).

25. *Line 150:Clarify terminology. To me, "carbon yield" refers to the mass of carbon measured, not the weight percent carbon in the dated material. Both are relevant: very small carbon masses can produce unreliable AMS results, while carbon abundance often scales with age offset.*

Authors' Response: We agree with the reviewer's comment. We have removed mention of carbon yield and replaced it with the term carbon content.

26. *Line 165:The effect of changing sedimentation rate on age offsets deserves further discussion. Consider the differing impacts of millennial-scale sedimentation trends (potentially a major effect) versus shorter-term (century-scale or shorter) fluctuations around a mean (little or no effect), as well as the unique challenges of estimating near-surface sedimentation rates, which are strongly influenced by porosity and core handling.*

Authors' Response: We thank the reviewer for this suggestion as he rightly comments that SAR are important in Baikal both from a spatial and temporal perspective. From a spatial perspective we emphasise that by far the majority of dates have been done on cores from underwater or isolated highs that experience hemipelagic sedimentation with minimal disruption from turbidite flows. (Line 210).

From a temporal perspective we have added greater justification as to why we focus on dates spanning the last 13 kyr.

- Autochthonous algae (especially diatoms and picoplankton) in Lake Baikal are by far the largest sources of organic material in Lake Baikal sediments. Like Carter and Colman (1994) our analysis is of dates younger than 13 $^{14}$C kyr BP, with relatively high concentrations of organic carbon being sequestered into the sedimentary record with onset of the warmer climate after the last glaciation. (Line 181)

- On the Academician Ridge, sedimentation rates are remarkably uniform of c. 4 cm/kyr for the past 250,000 years. Sedimentation rates are faster on the Buguldeika Saddle (between 15 - 19 cm/kyr) and again quite uniform for the past 30,000 years, due to the continuous mostly hemipelagic origin (Colman et al., 2003). However, given the resolution that most Baikal studies have been undertaken, centennial - decadal scales are unlikely to be captured (except for a few notable exceptions).

The reviewer is also right to highlight challenges to calculating near-surface sedimentation rates, which are strongly influenced by porosity and core handling. However, we have robust evidence from 210Pb analyses that cores taken using gravity or box corers preserve the surface sediment water interface (e.g. Morley et al., 2005; Swann et al., 2020). This is one of the main reasons why depths are able to be corrected, as detailed in our response to Point #23 above. When piston cores often lose their upper sediments; different coring methods used at the same location allow depth corrections to be made to sedimentary records.

27. *Line 168:I agree that assuming a constant age offset is a major weakness when using TOC for core chronologies. I suggest including estimates from the literature that document shifts in age offsets, and discussing how such variability could influence intercept ages.*

Authors' Response: We have added the following lines about temporal variability of age offsets to discuss how it could influence the intercept ages: "This may contribute to the spread in ASA estimates, alongside other sources of variability such as: temporal variability of sedimentation rate; temporal variability of age offset; and variable loss of top sediment during coring. Temporal variability of sedimentation rate or age offset will increase scatter in the results but are not expected to introduce a systematic bias.". (Line 515)

28. *Line 180:Clarify "median depth of those ages." Does this mean the median of all depths, or the midpoint of each sample?*

Authors' Response: Thank you for highlighting this. We mean the midpoint of each sample and have made this clearer in the manuscript (Line 188).

29. *Line 181:Using 14C ages rather than calibrated ages is reasonable given other potentially more important uncertainties. However, the rationale that the ages first need to be corrected before they are calibrated is weak. Calibrated ages can be used to estimate age offsets prior to correction. However, intercept (offset) values derived this way would be somewhat too low, as the slope of the age–depth regression would be steeper. The best estimate is probably somewhere between intercepts derived from 14C ages (too high) and from calibrated ages assuming no offset (too low). Presumably, code could be developed to iterate on the optimal intercept value.*

Authors' Response: We agree that there is room for improvement regarding the linear regression technique which may come from finding a way to use calibrated ages. We have removed this rationale.

30. *Line 197:Specify that the reported ages are based on total organic carbon.*

Authors' Response: Done.

31. *Line 197:Clarify what is meant by "mean" radiocarbon age. Mean of how many replicates?*

Authors' Response: Here we mean the "mean of the uncalibrated radiocarbon age's distribution". This has been amended to say "the mean of each (uncalibrated) radiocarbon age" (Line 171).

32. *Line 240:I disagree with the statement that "POM" and "FOM" represent "two different styles of sampling TOC." TOC includes molecular and other forms of organic carbon not retained by sieving, unlike POM and FOM.*

Authors' Response: The text has been amended to remove the suggestion that POM and FOM are styles of TOC. It has been replaced with: "POM and FOM relate to two different forms of organic matter, described by Colman et al. (1996): It was concluded that they were not statistically different to the TOC ages they reported.". (Line 243)

33. *Line 453:I disagree with the claim that "diatomaceous sediment cannot be used to infer the age offset for TOC." Colman et al. (1996) state that Lake Baikal sediments are diatomaceous muds during interglacial intervals.*

Authors' Response: We agree, and this statement has been deleted. Originally, we were referring to the fact that it was not clear that this sediment had been treated equivalently to TOC samples.

34. *Line 455:The key issue with using surface and trapped sediments to estimate age offsets is that they may contain bomb carbon, which could lead to underestimating the offset.*

Authors' Response: We thank the reviewer for this point and we have included in our text (Line 451)

35. *Line 481:Water residence time is directly related to reservoir age. I believe the authors mean "age offset." A long residence time can cause apparently old ages in plants assimilating dissolved carbon.*

Authors' Response: The reservoir age is related to whether the radiocarbon in a reservoir is in equilibrium with the radiocarbon in the atmosphere - the greater the disequilibrium the greater the reservoir age. The water residence time, which is the average time water takes to pass from inflow to outflow of the lake, has no impact on whether the radiocarbon in the water is in equilibrium with the radiocarbon in the atmosphere. The ventilation age, however, which describes how long, on average, water goes between successive ventilation (during which the water regains equilibrium with the atmosphere) does have an impact on reservoir age. We maintain that the water residence time is not itself mechanistically related to the reservoir age and so should not be used to estimate the reservoir age, as was done by (Nara et al., 2023).

36. *Line 495:"Identical" is too strong. Consider "indistinguishable" or a similar term.*

Authors' Response: Agreed, text has been amended to "statistically indistinguishable". (Line 509)

37. *Line 519:Be precise. For example: "an estimated 1σ uncertainty of ± 0.5 kyr."*

Authors' Response: We have highlighted that these are standard deviations.

38. *Line 519:The suggestion that the primary source of uncertainty in the offset is "error in the radiocarbon dates themselves" seems unlikely to me. The average error for 14C ages <13 ka in the dataset is ± 65 years. A more plausible source of uncertainty is temporal variability in offsets, long-term trend in sedimentation rate, and challenges in recovering undisturbed sediment–water interfaces. If sources of "dating error" other than analytical precision are suspected, they should be explained.*

Authors' Response: On reconsideration we agree with the reviewer. We have removed this sentiment and have added some discussion of sources of variability starting from line 526.

39. *Punctuation errors: Lines 91, 105, 235, 383, 517.*

Authors' Response: Amended.

*40. Typos :Lines 219, 312.*

Authors' Response: Amended

*41. Reference citation style: Many inconsistencies.*

Authors' Response: We have made sure that all citations use the journal style.

*42. Data check:Verify the reported 14C age of –13.365 ka (NUTA-3372).*

Authors' Response: We double checked; this is indeed what was reported by (Nakamura et al., 2003).

**Reviewer 2 Comments**

*1. With this study a database of all radiocarbon dates (published until 2023 and unpublished but accessible) from Lake Baikal is provided in a standardized report format and online accessible. This database is used to produce a new estimate for the age offset of TOC (does this TOC equal bulk organic matter?) in the sediments of Lake Baikal based on the largest possible sample size. As a result, it is concluded that the different depositional basins of Lake Baikal have no differences in age offset as suggested by previous publications. Instead, the temporal uncertainty of radiocarbon dates has been underestimated. This example for Lake Baikal points to the inherent uncertainty, which needs to be considered also for other lake records. Moreover, the authors request that consistent core labelling is important as well as comprehensive depth information. However, real "insights into ...radiocarbon age offsets" like documented in the title are not explicitly elaborated.*

Authors' Response: We thank the reviewer for these comments. One quick clarification first. Yes, TOC does here refer to bulk organic matter - the Baikal literature seems to prefer the term TOC so we stick to that here. We have changed the title to remove suggestion of "insights" and highlight that the paper's contribution is the creation of a database and an assessment of age offsets.

*2. The topic is interesting, stimulates new investigations and certainly suitable for "Earth System Science Data". The text is well written although there are plenty punctuation errors throughout. The structure is appropriate.*

Authors' Response: We have amended all the minor errors, including the many related to punctuation and different citation styles.

*3. Line (L) 24: during the entire abstract 14C yr BP are used as a unit. Only here, there is a shift to 14C kyr BP? For means of consistency, I suggest to use the same unit throughout. This might even go further, and perhaps the authors are inclined to change the entire text to one unit. At least it is unclear to the reader, why two different units (yr vs kyr) are in use.*

Authors' Response: Thank you for picking up on this, we have edited the manuscript to only use 14C kyr BP.

*4. L 219: the reference is lacking after the second "by".*
*L 235: this line should end with a period.*
*L 247: Instead "The top two rows…" it should run: "The top seven rows…" to correctly relate to the figure.*
*L 280: the second "was" needs to be deleted.*
*L 285: please avoid colloquial English! Same for L 437 and L 446.*
*L 517: delete the period.*

Authors' Response: Thank you for picking up on these. These minor corrections have all been made.

Björck, S. and Wohlfarth, B.: 14C Chronostratigraphic Techniques in Paleolimnology, in: Tracking Environmental Change Using Lake Sediments, vol. 1, edited by: Last, W. M. and Smol, J. P., Kluwer Academic Publishers, Dordrecht, 205–245, https://doi.org/10.1007/0-306-47669-X_10, 2002.

Carter, S. J. and Colman, S. M.: Biogenic Silica in Lake Baikal Sediments: Results From 1990–1992 American Cores, Journal of Great Lakes Research, 20, 751–760, https://doi.org/10.1016/S0380-1330(94)71192-8, 1994.

Colman, S. M., Jones, G. A., Rubin, M., King, J. W., Peck, J. A., and Orem, W. H.: AMS radiocarbon analyses from Lake Baikal, Siberia: Challanges of dating sediments from a large, oligotrophic lake, Quaternary Science Reviews, 15, 669–684, https://doi.org/10.1016/0277-3791(96)00027-3, 1996.

Colman, S. M., Karabanov, E. B., and Nelson, C. H.: Quaternary Sedimentation and Subsidence History of Lake Baikal, Siberia, Based on Seismic Stratigraphy and Coring, Journal of Sedimentary Research, 73, 941–956, https://doi.org/10.1306/041703730941, 2003.

Khider, D., Emile-Geay, J., McKay, N. P., Gil, Y., Garijo, D., Ratnakar, V., Alonso-Garcia, M., Bertrand, S., Bothe, O., Brewer, P., Bunn, A., Chevalier, M., Comas-Bru, L., Csank, A., Dassié, E., DeLong, K., Felis, T., Francus, P., Frappier, A., Gray, W., Goring, S., Jonkers, L., Kahle, M., Kaufman, D., Kehrwald, N. M., Martrat, B., McGregor, H., Richey, J., Schmittner, A., Scroxton, N., Sutherland, E., Thirumalai, K., Allen, K., Arnaud, F., Axford, Y., Barrows, T., Bazin, L., Pilaar Birch, S. E., Bradley, E., Bregy, J., Capron, E., Cartapanis, O., Chiang, H. -W., Cobb, K. M., Debret, M., Dommain, R., Du, J., Dyez, K., Emerick, S., Erb, M. P., Falster, G., Finsinger, W., Fortier, D., Gauthier, N., George, S., Grimm, E., Hertzberg, J., Hibbert, F., Hillman, A., Hobbs, W., Huber, M., Hughes, A. L. C., Jaccard, S., Ruan, J., Kienast, M., Konecky, B., Le Roux, G., Lyubchich, V., Novello, V. F., Olaka, L., Partin, J. W., Pearce, C., Phipps, S. J., Pignol, C., Piotrowska, N., Poli, M. -S., Prokopenko, A., Schwanck, F., Stepanek, C., Swann, G. E. A., Telford, R., Thomas, E., Thomas, Z., Truebe, S., Von Gunten, L., Waite, A., Weitzel, N., Wilhelm, B., Williams, J., Williams, J. J., Winstrup, M., Zhao, N., and Zhou, Y.: PaCTS 1.0: A Crowdsourced Reporting Standard for Paleoclimate Data, Paleoceanog and Paleoclimatol, 34, 1570–1596, https://doi.org/10.1029/2019PA003632, 2019.

Millard, A. R.: CONVENTIONS FOR REPORTING RADIOCARBON DETERMINATIONS, Radiocarbon, 56, 555–559, https://doi.org/10.2458/56.17455, 2014.

Morley, D. W., Leng, M. J., Mackay, A. W., and Sloane, H. J.: Late glacial and Holocene environmental change in the Lake Baikal region documented by oxygen isotopes from diatom silica, Global and Planetary Change, 46, 221–233, https://doi.org/10.1016/j.gloplacha.2004.09.018, 2005.

Nakamura, T., Oda, T., Tanaka, A., and Horiuchi, K.: High precision 14C age estimation of bottom sediments of Lake Baikal and Lake Hovsgol by AMS, Gekkan Chikyu, Special No.42, Kaiyoshuppasha, Tokyo, 20–31, 2003.

Nara, F. W., Watanabe, T., Lougheed, B. C., and Obrochta, S.: ALTERNATIVE RADIOCARBON AGE-DEPTH MODEL FROM LAKE BAIKAL SEDIMENT: IMPLICATION FOR PAST HYDROLOGICAL CHANGES FOR LAST GLACIAL TO THE HOLOCENE, Radiocarbon, 1–18, https://doi.org/10.1017/RDC.2023.63, 2023.

Philippsen, B.: The freshwater reservoir effect in radiocarbon dating, Herit Sci, 1, 24, https://doi.org/10.1186/2050-7445-1-24, 2013.

Stuiver, M. and Polach, H. A.: Discussion Reporting of $^{14}$C Data, Radiocarbon, 19, 355–363, https://doi.org/10.1017/S0033822200003672, 1977.

Watanabe, T., Nakamura, T., and Kawai, T.: Radiocarbon dating of sediments from large continental lakes (Lakes Baikal, Hovsgol and Erhel), Nuclear Instruments and Methods in Physics Research Section B: Beam Interactions with Materials and Atoms, 259, 565–570, https://doi.org/10.1016/j.nimb.2007.01.200, 2007.

Watanabe, T., Nakamura, T., Nara, F. W., Kakegawa, T., Nishimura, M., Shimokawara, M., Matsunaka, T., Senda, R., and Kawai, T.: A new age model for the sediment cores from Academician ridge (Lake Baikal) based on hightime-resolution AMS 14C data sets over the last 30 kyr: Paleoclimatic and environmental implications, Earth and Planetary Science Letters, 286, 347–354, https://doi.org/10.1016/j.epsl.2009.06.046, 2009a.

Watanabe, T., Nakamura, T., Nara, F. W., Kakegawa, T., Horiuchi, K., Senda, R., Oda, T., Nishimura, M., Matsumoto, G. I., and Kawai, T.: High-time resolution AMS 14C data sets for Lake Baikal and Lake Hovsgol sediment cores: Changes in radiocarbon age and sedimentation rates during the transition from the last glacial to the Holocene, Quaternary International, 205, 12–20, https://doi.org/10.1016/j.quaint.2009.02.002, 2009b.